# Maternal Death by COVID-19 Associated with Elevated Troponin T Levels

**DOI:** 10.3390/v14020271

**Published:** 2022-01-28

**Authors:** Johnatan Torres-Torres, Raigam Jafet Martinez-Portilla, Salvador Espino y Sosa, Juan Mario Solis-Paredes, Jose Antonio Hernández-Pacheco, Paloma Mateu-Rogell, Anette Cravioto-Sapien, Adolfo Zamora-Madrazo, Guadalupe Estrada-Gutierrez, Miguel Angel Nares-Torices, Norma Patricia Becerra-Navarro, Virginia Medina-Jimenez, Jose Rafael Villafan-Bernal, Lourdes Rojas-Zepeda, Diana Hipolita Loya-Diaz, Manuel Casillas-Barrera

**Affiliations:** 1Clinical Research Branch, Instituto Nacional de Perinatología Isidro Espinosa de los Reyes, Mexico City 11000, Mexico; torresmmf@gmail.com (J.T.-T.); raifet@hotmail.com (R.J.M.-P.); juan.mario.sp@gmail.com (J.M.S.-P.); antonhernp@yahoo.com.mx (J.A.H.-P.); dramateurogell@gmail.com (P.M.-R.); cravioto.anette@gmail.com (A.C.-S.); adolfozm84@gmail.com (A.Z.-M.); gpestrad@gmail.com (G.E.-G.); drnarestorices@hotmail.com (M.A.N.-T.); 2ABC Medical Center, Medical Association, Mexico City 05330, Mexico; npatriciabn@hotmail.com; 3Iberoamerican Research Network in Obstetrics, Gynecology and Translational Medicine, Mexico City 06720, Mexico; dravirginiamedina@gmail.com (V.M.-J.); joravibe@gmail.com (J.R.V.-B.); 4Laboratory of Immunogenomics and Metabolic Diseases, INMEGEN, Mexico City 14610, Mexico; 5Instituto Materno Infantil del Estado de México, Toluca 50170, Mexico; dra.rojaszepeda@gmail.com; 6Maternal Fetal Medicine Department, Hospital General de Mexico “Dr. Eduardo Liceaga”, Mexico City 06720, Mexico; dianitaldiaz@gmail.com; 7Hospital de la Mujer, Mexico City 11340, Mexico; mcasillasbarrera@gmail.com

**Keywords:** COVID-19, maternal death, troponin T

## Abstract

Cardiomyocyte injury and troponin T elevation has been reported within COVID-19 patients and are associated with a worse prognosis. Limited data report this association among COVID-19 pregnant patients. Objective: We aimed to analyze the association between troponin T levels in severe COVID-19 pregnant women and risk of viral sepsis, intensive care unit (ICU) admission, or maternal death. Methods: We performed a prospective cohort of all obstetrics emergency admissions from a Mexican National Institute. All pregnant women diagnosed by reverse transcription-polymerase chain reaction (RT-qPCR) for SARS-CoV-2 infection between October 2020 and May 2021 were included. Clinical data were collected, and routine blood samples were obtained at hospital admission. Seric troponin T was measured at admission. Results: From 87 included patients, 31 (35.63%) had severe COVID-19 pneumonia, and 6 (6.89%) maternal deaths. ROC showed a significant relationship between troponin T and maternal death (AUC 0.979, CI 0.500–1.000). At a cutoff point of 7 ng/mL the detection rate for severe pneumonia was 83.3% (95%CI: 0.500–0.100) at 10% false-positive rate. Conclusion: COVID-19 pregnant women with elevated levels of troponin T present a higher risk of death and severe pneumonia.

## 1. Introduction

COVID-19 is currently the leading cause of maternal death in Mexico; during 2021, it was responsible for 46% of maternal deaths, double the maternal mortality ratio compared with the pre-pandemic stage (31.1 vs. 54.5 per 100 thousand births) [1]. Pregnant women are more susceptible to severe pneumonia and death due to physiological changes of pregnancy and systemic inflammation induced by the SARS-CoV2 infection [2,3,4]. In addition, the high prevalence of comorbidities such as obesity, diabetes, hypertension, and chronic kidney disease are major risk factors for death in pregnant women with COVID-19 [5].

One of the main characteristics of COVID-19 is its capacity to evolve into a severe disease affecting multiple organs, including the endothelium and the heart [6]. Myocardiocyte damage due to SARS-CoV-2 is frequent and has been described in up to 8–20% of infected patients, particularly in severe forms of the disease [7]. It has also been described that patients infected with SARS-CoV-2 may develop cardiac injury with long-time consequences that may even require cardiac rehabilitation, leading to cardiac dysfunction and arrhythmias [8]. Several mechanisms could explain how SARS-CoV-2 infection affects the myocardium and how a clinical heart disease may be expected from severe COVID-19 [8]. SARS-CoV-2 can directly injure myocardial cells by inducing a cytokine rush resulting in myocardial oxygen (supply/demand). This mechanism has been described as a common way of myocardial injury in other diseases related to inflammatory response syndrome [7]. This hypothesis is supported by increased serum cardiac enzymes such as troponin I, troponin T, and bNP, which are logical markers of a worse prognosis in COVID-19 [9,10].

There are scarce reports on the association between cardiac enzymes (troponin I, troponin T, and bNP) with severe pneumonia, viral sepsis, ICU admission, and death in pregnant women. We hypothesized that cardiac enzymes may be elevated in pregnant women with COVID-19, compared with those without COVID-19, and that they could also be associated with severe clinical outcomes among women with COVID-19. Thus, this study aimed to evaluate the association between troponin T and the risk of severe adverse maternal outcomes in pregnant women with COVID-19.

## 2. Materials and Methods

### 2.1. Study Design and Participants

We conducted a prospective cohort study in the National Institute of Perinatology “Isidro Espinosa de los Reyes” and General Hospital of Mexico “Dr. Eduardo Liceaga”, in Mexico City. All symptomatic pregnant women with positive SARS-CoV-2 tests were included between December 2020 and September 2021. The Ethics and Research Internal Review Board of the National Institute of Perinatology approved the protocol (2020-1-32). All enrolled women signed informed consent.

### 2.2. Data Collection

The criterion for performing PCR was the identification of suggestive symptoms in the evaluation of emergency services, and blood samples were taken upon admission without taking into account the days of evolution of the symptoms. Medical data such as age, gestational age, pregestational body mass index (pBMI (kg/m^2^)), the status of chronic hypertension, pre-gestational diabetes, chronic renal disease, mean arterial pressure (MAP), oxygen saturation (SpO2), preeclampsia, threatened preterm labor, fetal growth restriction, stillbirth, pneumonia, viral sepsis, and mortality were collected from medical records. Rutinary blood samples were obtained at hospital admission. Troponin T and D-dimer serum levels were measured by an automated analyzer (Cobas-t511, Roche^®^, Mexico City, Mexico), whereas C-reactive protein serum levels were measured using an automated analyzer (Cobas-501, Roche^®^, Mexico City, Mexico) according to the manufacturer’s instructions. 

### 2.3. Outcome

The primary outcome was maternal death as a direct cause of COVID-19. Secondary outcomes were severe pneumonia, the requirement of ICU admission, and viral sepsis. 

Severe pneumonia was defined according to the American Thoracic Society criteria, which include either one major criterion (septic shock with the need for vasopressors or respiratory failure requiring mechanical ventilation) or three or more minor criteria (respiratory rate ≥ 30 breaths/min; PaO_2_/FIO_2_ ratio ≤ 250; multilobar infiltrates; confusion/disorientation; uremia (blood urea nitrogen level ≥ 20 mg/dL); leukopenia (white blood cell count < 4000 cells/µL); thrombocytopenia (platelet count < 100,000/µL); hypothermia (core temperature < 36 °C); hypotension requiring aggressive fluid resuscitation) [11,12]. ICU admission was decided according to the Quick Sequential Organ Failure Assessment (qSOFA) score, where a score ≥ 2 points would require ICU admission [13]. Viral sepsis is defined as life-threatening organ dysfunction caused by a dysregulated host response to infection, following the Third International Consensus Definitions for Sepsis and Septic Shock (Sepsis-3), organ dysfunction can be identified as an acute change in total SOFA score ≥2 points consequent to the confirmed SARS-CoV-2 infection [14].

### 2.4. Statistical Analysis

Descriptive and inferential statistics were used. Quantitative variables were reported as the median and interquartile range (IQR), while qualitative data were reported as numbers and percentages. Among patients with pneumonia, differences between variables were compared with maternal death using the Mann–Whitney U test or X^2^ test. Forward and backward stepwise logistic regression analyses were performed to assess the association between independent variables and the primary and secondary outcomes. After logistic regression, the adjusted model’s performance was evaluated by receiver-operating curve (ROC) analysis estimating the area under the curve (AUC). A *p*-value < 0.05 was considered significant. (StataCorp, 2020, Stata Statistical Software: Release 17; StataCorp LLC., College Station, TX, USA).

## 3. Results

### 3.1. Description of the Cohort 

A total of 115 pregnant women with suspected SARS-CoV-2 infection were included in the original cohort. In total, 28 were excluded because although they were COVID positive, they did not warrant hospital admission or additional biochemical evaluation, as they were asymptomatic. Consequently, 87 symptomatic pregnant women with SARS-CoV-2 infection were included in the statistical analysis: of those, 31 (35.63%) had severe COVID-19 pneumonia, including 6 (6.89%) maternal deaths. 

### 3.2. Comparison of Women with Severe and Non-Severe Pneumonia by COVID-19

In baseline characteristics, there were some differences between women with severe and non-severe pneumonia by COVID-19. Women with severe pneumonia had lower median gestational age at hospital admission (30.3 vs. 35.3; *p* = 0.002) and lower oxygen saturation (O2Sat%) (91.1 vs. 95.6; *p* = 0.005) than those with non-severe pneumonia. Women with severe pneumonia had significantly higher troponin T serum levels and other hematological and biochemical parameters than non-severe pneumonia. There was no significant correlation between the results of D-dimer with severity. Compared with non-severe pneumonia, those patients with severe pneumonia had a higher frequency of fetal growth restriction, stillbirth, ICU admission, viral sepsis, multiple organ dysfunction, and maternal death (Table 1).

### 3.3. Clinical and Biochemical Profile of Deceased Patients and Survivors

Among 31 pregnant women with severe pneumonia by COVID-19, there were six maternal deaths. Compared with survivors, the patients who died exhibit higher frequency of smoking habit, stillbirth (33.3% vs. 4%; *p* = 0.029), ICU admission (100% vs. 50%; *p* = 0.025), and viral sepsis (66.6% vs. 8%; *p* = 0.001). Additionally, they had higher levels of troponin T (*p* = 0.001) (70.45 vs. 92.25.; *p* = 0.009) and myoglobin (Table 1).

### 3.4. Association of Troponin and Maternal Outcome

Compared with non-severe COVID-19 (1.2 ng/mL), the median of troponin T serum levels in patients required ICU admission (5.7 ng/mL), with viral sepsis (12.3 ng/mL) and deceased (17.8 ng/mL) were significantly higher (Figure 1).

In ROC analysis, (n = 87) the troponin T predicted maternal death (AUC 0.833, CI 0.500–1.000). At a 10% false-positive rate, a cutoff point of 7 ng/mL predicted maternal death with a sensibility of 83.3% (Figure 2), which means that troponin T had an excellent predictive value for maternal death in the pregnant population with PCR positive for SARS-CoV-2 who presented symptomatically to the emergency department. This adequate balance between sensitivity and specificity for the analyzed outcomes motivated us to explore its predictive performance.

Through a multinomial logistic regression, we further analyzed the relationship of elevated troponin T with maternal outcomes. A troponin T value higher than 7 ng/mL was significantly related to severe pneumonia (OR 1.51, CI95% 1.15–1.98, *p* < 0.003), viral sepsis (OR 1.12, CI95% 1.008–1.254, *p* = 0.035), ICU admission (OR 1.17, CI95% 1.054–1.311, *p* = 0.004), and maternal death (OR 1.42, CI95% 1.13–1.784, *p* = 0.003) (Table 2).

## 4. Discussion

### 4.1. Main Findings

The principal findings of this study are that higher levels of troponin in pregnant women with severe pneumonia by COVID-19 are associated with a 1.17-fold increase in ICU admission, a 1.12-fold increase in viral sepsis, and a 1.42-fold increase in maternal death.

### 4.2. Comparison with Existing Literature

Cardiac troponins are used in clinical practice to study cardiac damage in acute coronary syndrome (ACS), septic shock, and more recently, in SARS-CoV-2 infection [15,16]. Elevated troponin levels are frequently found in patients with COVID-19 as a result of viral myocarditis, cytokine-driven myocardial damage, microangiopathy, and unmasked coronary artery disease [17].

Ruiz Mercedes et al. reported the elevation of serum troponins resulting from myocardial injury and left ventricular dysfunction in a case series of 15 pregnant women diagnosed with COVID-19, with a 13.3% mortality rate [18]. However, they did not study the relationship of troponin T with adverse maternal outcomes. Cortes-Telles et al. conducted a cohort of 200 Mexican patients where troponin T elevation was associated with an OR for mortality of 6.3 (CI95% 3.30–12.05) *p* < 0.001 [19]. Thus, our results in pregnant women with severe pneumonia reinforce the utility of troponin T as a prognostic marker related to death in pregnant women testing positive for COVID-19. In addition, levels higher than 7 ng/mL are related to the risk of sepsis, severe pneumonia, and admission to ICU.

Our results are supported by observations in non-pregnant critical COVID-19 patients for whom cardiac biomarkers (troponin I, troponin T, creatine kinase-MB, and myoglobin) predict poor prognosis [20].

Cardiomyocytes infection by SARS-CoV2 occurs through ACE 2 receptors, which are highly expressed in these cells. Under normal circumstances, ACE2 activity confers cardiovascular protection [21], because its peptidase activity cleaves angiotensin II (Ang II) into angiotensin 1–7 (Ang 1–7) [22,23]. As a result of infection, ACE2 activity decreases because it binds to SARS-CoV-2, and the complex is internalized to the cell. [24] In consequence, ACE activity is increased, producing an elevation of Ang II serum levels, which promotes cardiac damage/injury through vasoconstriction [21].

In those diagnosed with COVID-19, Ang II serum levels are lower in deceased patients, compared with surviving patients [25]. A hypothetical mechanism leading to this relative decrease in Ang II levels in non-surviving patients is through a greater binding of Ang II to cell surface AT1 (AT1a) receptors, which induces downstream signaling responses, followed by quicker endocytosis of the Ang II/AT1. Excess cytoplasmic Ang II induces overproduction of mitochondrial reactive oxygen species (ROS) [26].

Oxidative damage of cardiomyocytes causes an alteration to the membrane integrity leaking cardiac troponin T. It also seems that COVID-19 reduces blood flow to coronary arteries resulting in myocardial damage. SARS-CoV-2 may lead to severe endothelial inflammation, leading to atherosclerotic plaque instability and rupture [27]. Aside from vasoconstriction and hypercoagulable state, all these mechanisms contribute to myocardial damage [28].

### 4.3. Strengths and Limitations 

Some limitations should be reported. Firstly, due to the span of the study, the sample was relatively small; therefore, analysis of a larger patient’s sample will be needed for external validation and replication in other populations. In addition, some other specific myocardial damage biomarkers such as troponin I and pro-BNP could not be measured, and therefore, an improved approach that includes the analysis of these cardiac enzymes is suggested in future research.

### 4.4. Clinical Interpretation

Decision-making can benefit from the evaluation of functional and/or structural cardiovascular damage through monitoring and surveillance with electrocardiography and echocardiography, screening of inflammatory and cardiac biomarkers, mainly troponin T, and inflammatory markers of the acute phase of the disease such as water balance, together with SOFA, APACHE, MEWT, MEOWS scores at hospital admission of all pregnant women with COVID-19.

From the fetal point of view, carditis is one of the components of multiple organ failure due to SARS-CoV-2, (which causes inflammation of the myocardium with troponin release) that leads to heart failure and arrhythmias, decreasing cardiac output, which affects the uteroplacental flow, and it would have repercussions on the efficiency of the exchange membrane; this causal relationship has not been studied. This supports fetal surveillance during the critical period of COVID-19 with an emphasis on characterizing the area of exchange and redistribution of fetal vascular flows.

## 5. Conclusions

Elevated levels of troponin T among pregnant women with COVID-19 present a higher risk of death. Therefore, myocardial biomarkers should be evaluated in pregnant patients with COVID-19 that require hospital admission for risk stratification purposes. 

## Figures and Tables

**Figure 1 viruses-14-00271-f001:**
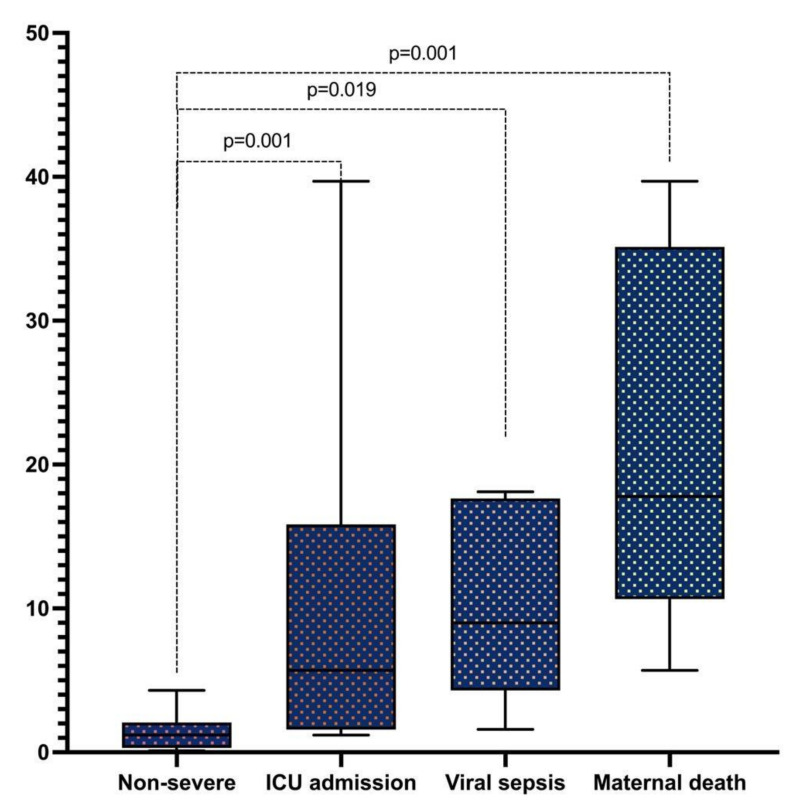
Troponin T levels in severe COVID-19 pregnant women with adverse outcomes. Non-severe n = 56 (1.2 ng/mL (0.3–2.05 ng/mL)); ICU admission n = 18 (5.7 ng/mL (1.25–14.9 ng/mL)); viral sepsis n = 6 (12.3 ng/mL (5.7–17.5 ng/mL)); maternal death n = 6 (17.8 ng/mL (12.3–33.6 ng/mL)).

**Figure 2 viruses-14-00271-f002:**
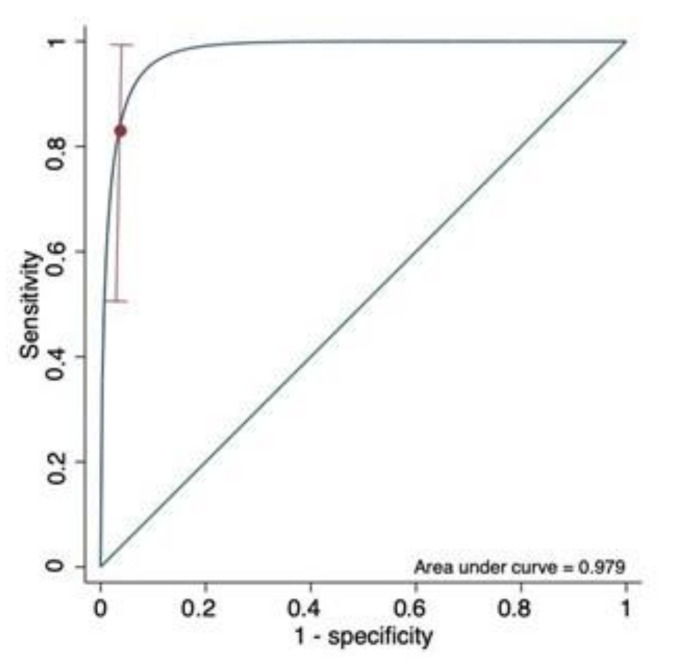
Area under the receiver-operating curve (ROC) of troponin T for the prediction of maternal death by COVID-19. Detection rate for maternal death at a 10% false-positive rate was 83.3% (95%CI:0.500–0.100).

**Table 1 viruses-14-00271-t001:** Clinical characteristics of the included population.

Characteristics	COVID-19		
	Non-Severe Pneumonia n = 56	Severe Pneumonia n = 31	*p*-Value
Maternal age	30.46 (26.38–33.47)	31.19 (26.38–35.55)	0.361
Gestational age at diagnosis	35.3 (30–39.1)	30.3 (25.1–33.3)	0.002
pBMI (kg/m^2^)	31.42 (26.7–34.25)	26.83 (25.39–33.05)	0.135
MAP (mmHg)	85.5 (80.16–92)	87.66 (80–95.33)	0.657
Smoking	0(0%)	2 (6.45%)	0.054
Chronic hypertension	2 (3.57%)	1 (3.33%)	0.954
Pre-gestational diabetes	3 (5.36%)	0(0%)	0.190
Chronic kidney disease	2 (3.57%)	0(0%)	0.287
SpO2%	95.6 (93–96.4)	91.1 (79.4–95.4)	0.005
Leukocytes (x10/L)	8.15 (7–10)	9.8 (7.8–13.5)	0.017
Neutrophils (x10/L)	6.35 (5.3–7.4)	8.8 (6.9–13)	0.0003
Glucose (mg/dL)	77.5 (73–85)	87 (75–116)	0.006
Troponin T (ng/mL)	1.2 (0.3–2.05)	2.7 (1.3–7.1)	0.0001
Myoglobin	17.8 (13.4–27.1)	29.3 (19.2–48)	0.010
D-dimer (ng/mL)	1916 (1262–2953)	1425 (1184–4083)	0.532
C-RP	16.25 (7.04–60.6)	110.5 (63.8–200)	0.0001
AST (U/L)	19 (15–32)	31 (24–49)	0.0001
ALT (U/L)	16 (11–28)	26 (18–40)	0.003
LDH (U/L)	173 (140–201)	249 (192–375)	0.0001
Direct bilirubin	0.1 (0.07–0.18)	0.22 (0.13–0.38)	0.001
Indirect bilirubin	0.32 (0.24–0.44)	0.42 (0.31–0.49)	0.075
Cholesterol	206.5 (171–240)	155 (128–187)	0.0003
Procalcitonin	0.05 (0.03–0.13)	0.39 (0.18–0.68)	0.0001
Preeclampsia (clinical diagnosis)	9 (16.36%)	8 (25.81%)	0.291
Threatened preterm labor	3 (5.56%)	3 (9.68%)	0.475
Fetal growth restriction	4 (7.41%)	7 (23.33%)	0.038
Stillbirth	0(0%)	3 (9.68%)	0.020
ICU admission	0(0%)	18 (60%)	<0.0001
Viral sepsis	0(0%)	6 (19.35%)	0.001
Multiple organ dysfunction	0(0%)	4 (12.9%)	0.006
Maternal death	0(0%)	6 (19.35%)	0.001

pBMI: pre-gestational body mass index; MAP: mean arterial pressure; MoM: multiples of the median; SpO2: oxygen saturation; Mann–Whitney U test for continuous variables is expressed as median and interquartile range; Χ^2^ or Fisher’s test for categorical variables is expressed as number and percentage.

**Table 2 viruses-14-00271-t002:** COVID-19 complications associated with elevated troponin.

	OR	95%CI	*p*-Value	aOR	95%CI	*p*-Value
Severe pneumonia	1.52	1.156–1.991	0.003	1.51	1.151–1.983	0.003
Viral sepsis	1.08	1.003–1.169	0.039	1.12	1.008–1.254	0.035
ICU admission	1.17	1.037–1.319	0.01	1.17	1.054–1.311	0.004
Maternal death	1.27	1.084–1.498	0.003	1.42	1.13–1.784	0.003

OR: odds ratio; aOR: adjusted odds ratio with BMI; ICU: intensive care unit.

## Data Availability

The data presented in this study are available on request from the corresponding author. The data are not publicly available due to privacy.

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
