# Peer review of "Maternal Death by COVID-19 Associated with Elevated Troponin T Levels"

_viruses, 2022, doi:10.3390/v14020271_

Round 1

Reviewer 1 Report

This article was a great and interesting work related to maternal death by COVID-19 and troponina T. However, some kindly suggestions would do to improve the manuscript

Abstract: In the methods, make the inclusion criteria clear. Inform that pregnant women were included

Introduction: This section is well written and provides important elements about myocardial involvement in COVID-19 and the markers of cardiac injury. However, it would also be interesting to include information about COVID-19 in pregnant women and the rate of maternal death in Mexico during the period.

Methods:

As the severity of COVID-19 is revealed with the evolution of the disease, could the authors inform on what day of the onset of symptoms the pregnant women were hospitalized and the tests collected?

Was the assessment of percentage of pulmonary involvement in pregnant women carried out through imaging tests? Could this data contribute to the study?

Could the authors better define viral sepsis, not just referring to the consensus?

Table 1 includes the result of the PI of the uterine artery. I believe this parameter is secondary and has little relevance for the present study. I suggest exclusion or better reasoning to analyze this data and inclusion of the description in the methods.

Results

As the study included 87 patients for analysis, I suggest including a table with the main baseline characteristics of the population in the main text, not just in the supplementary material.

Table 1: Exclude “UtAPI: uterine artery pulsatility índex” from the footnote and include the meaning of “IpmAut”.

In topic 3.4, it is not clear whether the analysis included 87 patients or only the 31 with pneumonia by COVID.

In ROC analysis, it is not clear the total of cases included. Please inform it.

Discussion

The authors need to clarify, why did they built models using the cut-off of 7 for Toponin T.

Did the authors perform repeated measurements of troponin T throughout the hospitalization of the pregnant woman? Was it possible to detect any relevant pattern that indicates maternal or fetal repercussions?

It is known that the uteroplacental flow depends on cardiac output, consequently maternal cardiac dysfunction will harm the pregnancy. In these situations, decision-making about childbirth is complex. Could the authors discuss more about the particularities of management in obstetric patients?

Finally, I encourage the work team to continue investigating this topic in order to establish intervention strategies that prevent maternal deaths.

Author Response

The research team is deeply grateful for your valuable observations and comments, we hope to give a full response to each of them and I remain pending any further indication to give a timely response.

  1. Abstract: In the methods, make the inclusion criteria clear. Inform that pregnant women were included

the original sentence was: All patients diagnosed by reverse transcription-polymerase chain reaction (RT-qPCR) for SARS-CoV-2 infection between October 2020 and May 2021 were included.

we specify that it is about pregnant women:

All pregnant women diagnosed by reverse transcription-polymerase chain reaction (RT-qPCR) for SARS-CoV-2 infection between October 2020 and May 2021 were included.

  1. Introduction: This section is well written and provides important elements about myocardial involvement in COVID-19 and the markers of cardiac injury. However, it would also be interesting to include information about COVID-19 in pregnant women and the rate of maternal death in Mexico during the period.

We included the following paragraph

During the year 2021, it was responsible for 46% of maternal deaths, doubling the maternal mortality ratio when compared to the pre-pandemic stage (31.1 vs. 54.5 per 100 thousand births).

Methods:

  1. As the severity of COVID-19 is revealed with the evolution of the disease, could the authors inform on what day of the onset of symptoms the pregnant women were hospitalized and the tests collected?

Since we do not establish a day of evolution of the symptoms for taking the tests and we consider that it is not very reproducible to establish the days of evolution according to the questioning, we include the following sentence:

The criterion for performing PCR was the identification of suggestive symptoms in the evaluation of the emergency service and blood samples were taken upon admission, without taking into account the days of evolution of the symptoms

  1. Was the assessment of percentage of pulmonary involvement in pregnant women carried out through imaging tests? Could this data contribute to the study?

We did not perform CT scans in the patients given the lack of availability of the resource and that it was not required as part of clinical decision-making during pregnancy complicated with COVID-19

  1. Could the authors better define viral sepsis, not just referring to the consensus?

We included the following paragraph:

Viral sepsis is defined as life-threatening organ dysfunction caused by a dysregulated host response to infection, in accordance with the Third International Consensus Definitions for Sepsis and Septic Shock (Sepsis-3), organ dysfunction can be identified as an acute change in total SOFA score ≥2 points consequent to the confirmed SARS-COV-2 infection.

  1. Table 1 includes the result of the PI of the uterine artery. I believe this parameter is secondary and has little relevance for the present study. I suggest exclusion or better reasoning to analyze this data and inclusion of the description in the methods.

we fully agree with the observation, we have suppressed the uterine artery pulsatility index

Results

  1. As the study included 87 patients for analysis, I suggest including a table with the main baseline characteristics of the population in the main text, not just in the supplementary material.

The table with the baseline characteristics was included.

  1. Table 1: Exclude “UtAPI: uterine artery pulsatility índex” from the footnote and include the meaning of “IpmAut”.

The UtAPI has been suppressed from de analysis

  1. In topic 3.4, it is not clear whether the analysis included 87 patients or only the 31 with pneumonia by COVID.

We have included in the paragraph the n of patients with non-severe COVID (56), the original:

Compared to non-severe COVID-19 (1.2 ng/ml), the median of troponin T serum levels in patients required ICU admission (5.7 ng/ml), with viral sepsis (12.3 ng/ml) and deceased (17.8 ng/ml) were significantly higher, the uterine artery pulsatility index between the study groups shows that poor prognosis was not associated with preeclampsia (Figure 1).

was changed to:

Compared to the 56 symptomatic non-severe COVID-19 (1.2 ng/ml), the median of troponin T serum levels in patients required ICU admission (5.7 ng/ml), with viral sepsis (12.3 ng/ml) and deceased (17.8 ng/ml) were significantly higher, the uterine artery pulsatility index between the study groups shows that poor prognosis was not associated with preeclampsia (Figure 1).

  1. In ROC analysis, it is not clear the total of cases included. Please inform it.

We include the n and add an interpretation to it for clearer reading.

In ROC analysis, (n=87) the troponin T predicted maternal death (AUC 0.833, CI 0.500-1.000). At a 10% false-positive rate, a cut-off point of 7 ng/mL predicted maternal death with a sensibility of 83.3% (Figure 2), which means that troponin T had an excellent predictive value for maternal death in the pregnant population with PCR positive for SARS-COV-2 who presented symptomatic to the emergency department.

Discussion

  1. The authors need to clarify, why did they build models using the cut-off of 7 for Troponin T.

The justification is in the last observation, so we add the following paragraph:

this adequate balance between sensitivity and specificity for the analyzed outcomes motivated us to explore its predictive performance

  1. Did the authors perform repeated measurements of troponin T throughout the hospitalization of the pregnant woman? Was it possible to detect any relevant pattern that indicates maternal or fetal repercussions?

Excellent observation! unfortunately, we have not performed sequential measurements of the markers. It is a question that we will have to ask in the cohort

  1. It is known that the uteroplacental flow depends on cardiac output, consequently maternal cardiac dysfunction will harm the pregnancy. In these situations, decision-making about childbirth is complex. Could the authors discuss more about the particularities of management in obstetric patients?

Thank you very much for this valuable observation. We add the following paragraph

From the fetal point of view, carditis is one of the components of multiple organ failure due to SARS-COV-2, (which causes inflammation of the myocardium with troponin release) that leads to heart failure and arrhythmias, decreasing cardiac output, which affects the uteroplacental flow and it would have repercussions on the efficiency of the exchange membrane, this causal relationship has not been studied. This supports fetal surveillance during the critical period of COVID-19 with an emphasis on characterizing the area of ​​exchange and redistribution of fetal vascular flows.

Finally, I encourage the work team to continue investigating this topic in order to establish intervention strategies that prevent maternal deaths.

The research team appreciates your time and professionalism, we work in translational research, we hope to soon validate intervention strategies in this group of patients, let me inform you that we have recently signed an opinion statement for the Mexican Federation of Obstetrics and Gynecology Colleges Regarding vaccination for COVID-19 in pregnancy, we hope it will have a national impact on public policy.

Reviewer 2 Report

The present study shows that the elevation of troponin T was associated with maternal death and severe pneumonia in COVID-19 pregnant women. I have several concerns as listed below.

  1. Troponin has been already reported that correlate with severe prognosis in COVID-19 patients and can be used as predictive biomarkers. Therefore, the novelty of the present manuscript is limited. The authors should refer to the previous article adequately, then discuss more in detail.
  2. Fig.1 The authors should describe the number of patients in each group in the legend.

Author Response

The research team is deeply grateful for your valuable observations and comments, we hope to give a full response to each of them and I remain pending any further indication to give a timely response.

  1. Troponin has been already reported that correlate with severe prognosis in COVID-19 patients and can be used as predictive biomarkers. Therefore, the novelty of the present manuscript is limited. The authors should refer to the previous article adequately, then discuss more in detail.

Troponin has been used as a prognostic marker for severe COVID-19 in non pregnant women. We evaluated the performance of troponin in pregnant women and found similar results as the one published in non-pregnant participants.

Millman M, Santos ABS, Pianca EG, Pellegrini JAS, Conci FC, Foppa M. Rapid prognostic stratification using Point of Care ultrasound in critically ill COVID patients: The role of epicardial fat thickness, myocardial injury and age. J Crit Care. 2022;67:33-38. doi:10.1016/j.jcrc.2021.09.013

2. Fig.1 The authors should describe the number of patients in each group in the legend.

we added the following data to the figure

Non-severe n=56 (1.2 ng/ml [0.3-2.05 ng/ml]); ICU admission n=18 (5.7 ng/ml [1.25-14.9 ng/ml]); Viral sepsis n=6 (12.3 ng/ml [5.7-17.5 ng/ml]); Maternal death n=6 (17.8 ng/ml [12.3-33.6 ng/ml]).

Round 2

Reviewer 2 Report

The authors revised the manuscript adequately.